# Improved Characterization of Circulating Tumor Cells and Cancer-Associated Fibroblasts in One-Tube Assay in Breast Cancer Patients Using Imaging Flow Cytometry

**DOI:** 10.3390/cancers15164169

**Published:** 2023-08-18

**Authors:** Anna Muchlińska, Robert Wenta, Wiktoria Ścińska, Aleksandra Markiewicz, Grażyna Suchodolska, Elżbieta Senkus, Anna J. Żaczek, Natalia Bednarz-Knoll

**Affiliations:** 1Laboratory of Translational Oncology, Intercollegiate Faculty of Biotechnology, University of Gdańsk and Medical University of Gdańsk, 80-211 Gdańsk, Poland; 2Department of Oncology and Radiotherapy, Medical University of Gdańsk, 80-211 Gdańsk, Poland

**Keywords:** breast cancer, liquid biopsy, circulating tumor cells, circulating cancer-associated fibroblasts, imaging flow cytometry

## Abstract

**Simple Summary:**

Liquid biopsy is a promising but challenging tool for potentially upgrading cancer patient diagnostics and providing new insights into tumor biology. Here, we applied a unique approach to detect CTCs and cCAFs in a one-tube assay using imaging flow cytometry, enabling improved enumeration, multimarker-based phenotyping, and detailed morphological characterization of those rare cells. We identified new, putatively EMT-related phenotypes of negative CTCs for both epithelial and mesenchymal markers and showed that EMT-related CTCs might contribute to breast cancer progression, whereas a coincidence of CTCs and cCAFs might be a signature of metastasis.

**Abstract:**

Circulating tumor cells (CTCs) and circulating cancer-associated fibroblasts (cCAFs) have been individually considered strong indicators of cancer progression. However, technical limitations have prevented their simultaneous analysis in the context of CTC phenotypes different from epithelial. This study aimed to analyze CTCs and cCAFs simultaneously in the peripheral blood of 210 breast cancer patients using DAPI/pan-keratin (K)/vimentin (V)/alpha-SMA/CD29/CD45/CD31 immunofluorescent staining and novel technology—imaging flow cytometry (imFC). Single and clustered CTCs of different sizes and phenotypes (i.e., epithelial phenotype K+/V− and epithelial–mesenchymal transition (EMT)-related CTCs, such as K+/V+, K−/V+, and K−/V−) were detected in 27.6% of the samples and correlated with metastases. EMT-related CTCs interacted more frequently with normal cells and tended to occur in patients with tumors progressing during therapy, while cCAFs coincided with CTCs (mainly K+/V− and K−/V−) in seven (3.3%) patients and seemed to correlate with the presence of metastases, particularly visceral ones. This study emphasizes the advantages of imFC in the field of liquid biopsy and highlights the importance of multimarker-based analysis of different subpopulations and phenotypes of cancer progression-related cells, i.e., CTCs and cCAFs. The co-detection of CTCs and cCAFs might improve the identification of patients at higher risk of progression and their monitoring during therapy.

## 1. Introduction

For the last two decades, liquid biopsy has been developed dynamically to provide solutions (i.e., ultrasensitive technologies and/or specific markers) relevant to the early diagnosis and monitoring of cancer patients [1]. Circulating tumor cells (CTCs), though very plastic and heterogenous due to, e.g., epithelial–mesenchymal transition (EMT) [2,3], have been shown to be a non-invasive, prognostic biomarker in many solid tumors, including breast cancer (BC), one of the most common cancers worldwide among the female population. Yet, other components released from primary or secondary tumors, such as circulating endothelial cells, cancer-associated macrophage-like cells, or fibroblasts (CAFs), as well as extracellular vesicles, can also be found in the blood of cancer patients and are putatively correlated with worse prognosis [4,5,6].

Key players in dissemination and, more importantly, in metastasis formation are naturally CTCs, the “seeds” of new lesions [7]. In concordance, the poor prognostic connotation of CTCs has been demonstrated in both early [8] and metastatic breast cancer [9] and many other solid tumors [10,11,12]. In particular, intermediate and full EMT-related phenotypes (i.e., epithelial–mesenchymal and mesenchymal phenotypes) have been identified as potentially more aggressive ones, and possibly even more importantly, difficult to detect using common epithelial marker-dependent methods [2,13]. Furthermore, it was shown in a mouse model of lung cancer metastases that CTCs can also carry stromal cells, including activated fibroblasts, as fragments of “soil” ready to promote metastasis formation [14]. In accordance with this model, cCAFs were observed in the blood of cancer patients both individually and clustered with CTCs [6,15,16]. Interestingly, cCAFs have been isolated mostly from the blood samples of metastatic cancer patients, whereas CTCs have been detected in the blood samples of patients at all stages of disease, including early cancers [6,15,17]. 

Of note, the co-detection of CTCs and cCAFs in a one-tube assay has been difficult as multimarker-based identification of rare cells in blood is still limited, even using the golden standard CellSearch. The exact co-detection of CTCs and cCAFs has been restricted exclusively to epithelial CTCs, missing information about potential EMT-related phenotypes of CTCs [6,16]. Meanwhile, new technologies, not relying on exclusively epithelial markers, such as imaging flow cytometry (imFC) [18], might enable the detailed analysis of different subpopulations of cells in a one-tube assay in a relatively short time. As imFC offers a combination of the features of classical flow cytometry (including an impartial multiparameter and high-throughput analysis) and high-resolution fluorescence microscopy (allowing insight into the morphology of each individual examined object), it might boost investigation into tumor dissemination.

Therefore, in the present study, the feasibility of imFC for simultaneous CTC and cCAF detection from the blood samples of breast cancer patients was evaluated. Furthermore, the clinical relevance of those rarely detected cells was assessed. 

## 2. Materials and Methods

### 2.1. Breast Cancer Patients and Non-Cancer Volunteer Cohort

Two hundred ten female breast cancer patients (age ≥ 18, with no coexisting malignancy) treated in the Breast Unit, at the University Clinical Center in Gdańsk between 2019 and 2022, and 20 female volunteers (age ≥ 18) with no cancer history (called healthy volunteers) from the Central Clinical Laboratory at the University Clinical Center in Gdańsk were recruited for this study after providing written informed consent. Their clinico-pathological parameters, pretreatment blood count (incl. leukocytes and their subtypes), and response to therapy were documented for the breast cancer patients (Section 3.3). This study was conducted in accordance with the Declaration of Helsinki and approved by the Independent Bioethics Committee for Scientific Research at the Medical University of Gdańsk (protocol no. NKBBN/748/2019–2020).

### 2.2. Blood Collection and Processing for CTC Enrichment

EDTA-peripheral blood samples (vol. of 7.5 mL) were collected from all recruited patients and processed as soon as possible after blood donation (with 68% of the samples processed within up to 3 h after donation). First, 3 mL of blood was discarded to avoid skin cell (i.e., keratinocytes, fibroblasts) and endothelial cell contamination during the puncture. The peripheral blood mononuclear cell (PBMC) fraction was isolated using density gradient centrifugation. Briefly, the blood was centrifuged at 200× *g* for 10 min at room temperature to separate the platelet-rich plasma. Then, the remainder of the blood sample was diluted with 1x phosphate-buffered saline (1xPBS) up to 9 mL, layered onto the Histopaque^®^-1077 (Sigma-Aldrich, St. Louis, MO, USA), and centrifuged at 400× *g* for 30 min with break off. The PBMC fraction was harvested, fixed in 4% formaldehyde, and stored at −80 °C in 0.5 mL aliquots for further analysis.

### 2.3. Cell Culture

Primary breast cancer cell lines: luminal A: MCF7 (HTB-22), T-47D (HTB-133), luminal B: BT-474 (HTB-20), HER2+: SKBR3 (HTB-30), TNBC: HCC1806 (CRL-2335), MDA-MB-231 (HTB-26); metastatic luminal B breast cancer cell line: MDA-MB-361 (HTB-27); and endothelial cell line HMEC-1 (CRL-3243) were purchased from the American Tissue Culture Collection (ATCC, Manassas, VA, USA) and used for immunofluorescent staining protocol optimization. The cells were cultured in Dulbecco’s Modified Eagle Medium (DMEM) supplemented with 10% fetal bovine serum (FBS) under appropriate conditions and routinely tested for mycoplasma contamination. For all cell lines, the same batch of FBS was used. The cells were cultured until 80% of confluence, trypsinized, fixed in 4% formaldehyde, and stored at −80 °C for further analysis.

### 2.4. Immunofluorescent Staining for Simultaneous CTC and cCAF Detection

Immunofluorescent staining of the different breast cancer cell lines and clinical samples was performed using a cocktail of antibodies, aiming to detect CTC markers: pan-keratins (K, AE1/AE3 clone, AF488-conjugated, Thermo Fisher Scientific, Waltham, MA, USA, #53-9003-82; C11 clone, AF488-conjugated, Thermo Fisher Scientific, #MA5-18156) and vimentin (V, D21H3 clone, AF647-conjugated, Cell Signaling, Danvers, MA, USA, #9856); cCAF markers: alpha smooth muscle actin (*α*-SMA, 1A4 clone, PE-conjugated, R&D Systems, Minneapolis, MN, USA, #IC1420P) and CD29 (TS2/16 clone, SuperBright600-conjugated, Thermo Fisher Scientific, #63-0299-42); leukocyte markers: CD45 (REA747 clone, APC-Vio770-conjugated, Miltenyi Biotec, North Rhine-Westphalia, Germany, #130-110-635); and endothelial cell markers: CD31 (WM59 clone, APC-Cy7-conjugated, BioLegend, San Diego, CA, USA, #303120), diluted 1:2500, 1:2500, 1:50, 1:100, 1:1000, 1:50, and 1:10, respectively (prepared in 1x Perm-Wash Buffer, BD Biosciences, Franklin Lakes, NJ, USA). For the optimization of the pan-keratin staining, the MCF7 breast cancer cell line was used; for CD31—endothelial cell line HMEC-1; and for CD45 and vimentin—PBMC, whereas for the optimization of *α*-SMA staining, we selected the breast cancer cell line Hs578t, which displayed a significant expression of *α*-SMA. The investigated cells were thawed and washed with 1 mL of 1x PBS to remove the formaldehyde, and then incubated for 30 min at +4 °C with an antibody cocktail. After one additional washing step in 1x PBS, the cells were resuspended in 30 μL of 1x PBS, counterstained with DAPI (BD Biosciences, 1 μg/mL), and immediately analyzed using the Amnis^®^ ImageStream^®^ X Mk II (Luminex, Austin, TX, USA).

### 2.5. Imaging Flow Cytometry Analysis

The Amnis^®^ ImageStream^®^ X Mk II (Luminex), equipped with lasers at 405 nm, 488 nm, and 642 nm and INSPIRE™ software (version 200.1.681.0; Luminex), was used for sample analysis and data acquisition. The subsequent analysis of the obtained outcomes was performed with IDEAS software (version 6.2; Luminex). The cells were imaged at 40× magnification at a low speed to obtain high-quality images. For the markers potentially coexisting in the same subpopulation of cells and detected using the same laser (i.e., V, CD45, CD31), and highly intense fluorescence in the DAPI channel, the optimal compensation matrix between the individual fluorescence channels was established using a mixture of MDA–MB–231 cells, MCF7 cells, and PBMCs from a healthy donor, stained with V, CD45, and CD31 antibodies and DAPI separately or in combination. The experiment was repeated three times, utilizing samples from different healthy donors to account for potential inter-individual variations in the fluorescence signals. The optimized parameter settings for acquisition were used for all analyzed samples (Appendix A). All analyzed cells were verified visually in each fluorescent channel based on the fluorescence intensity and pattern to exclude autofluorescence and/or fluorescence crosstalk. Only distinct fluorescence patterns against the background were treated as true events.

All DAPI+ cells were counted and considered as the initial gate, whereas DAPI+/CD45−CD31− cells were collected. To gate potential CTCs, the fluorescence intensities of K and V were visualized in a 2D dot plot. To gate cCAFs, the fluorescence intensities of *α*-SMA and CD29 were visualized in histograms and cut at 10^4^ (Appendix A). The potential preselected CTCs and cCAFs were visually confirmed from immunofluorescence images for their morphology and staining details, tagged, and counted manually.

The detected cells were defined as: epithelial CTCs if K+/DAPI+/V−/CD45−CD31− (abbreviated to K+V−), mesenchymal CTCs if K−/DAPI+/V+/CD45−CD31− (K−V+), epithelial–mesenchymal CTCs if K+/DAPI+/V+/CD45−CD31− (K+V+), negative for both epithelial and mesenchymal markers if K−/DAPI+/V−/CD45−CD31− (K−V−), and cCAFs if *α*-SMA+/K−/DAPI+/V±/CD45−CD31− or CD29+/K−/DAPI+/V±/CD45−CD31− phenotypes were detected. The patients were classified according to their detected CTC phenotype into two classification systems. The exclusive phenotype was defined as the only phenotype found in the blood of an individual patient; otherwise, the patient was assigned to the “heterogenous CTCs” group. The dominant phenotype was assigned as the phenotype most enriched in the individual patient over the other coexisting phenotypes, whereas if some phenotypes were equally numerous, the “heterogenous CTCs” group was assigned. The CTC dimensions, such as the diameter, circumference, area, and circularity, were measured using QuPath ver. 0.2.3 software [19]. Cytoplasmatic protrusions were defined as fragments of the cell membrane extending beyond the estimated overall cell geometry.

To detect CTC clusters, the DAPI+ cell area was visualized in a histogram, and objects bigger than single CTCs, exhibiting a cytoplasm that could be segmented into at least 2 cells, along with a minimum of 2 visibly separated nuclei, were manually tagged and counted.

The presence, numbers, clusters, and phenotypes of CTCs or cCAFs, as well as their morphological details and interactions, were further correlated with the clinico-pathological data (i.e., age, T, N, and M status, grading, hormone receptor and HER2 status, molecular subtype, pretreatment blood count, and response to therapy).

### 2.6. Statistics

Statistical analysis was performed with the SPSS 27.0.1.0 software package (SPSS Inc., Chicago, IL, USA) licensed for the University of Gdańsk. The exact numbers of the detected CTCs and cCAFs were calculated per 1 million PBMCs to normalize all samples. The different classifications of the patients in regard to their CTC status were investigated, including the exclusive CTC phenotype, dominant CTC phenotype, and epithelial vs. EMT-related phenotype assigned to an individual patient. Chi-squared, Fisher’s exact, and Fisher–Freeman–Haltman tests were used to compare CTC- and/or cCAF-positive or negative patients and patients with different CTC phenotypes with clinico-pathological parameters (age, T stage, N stage, M stage, grade, molecular subtype, and response to treatment). Differences between the enumeration of CTCs and cCAFs in the BC patient group with different clinico-pathological parameters were assessed by Mann–Whitney and Kruskal–Wallis tests. All statistical analyses were two-sided, and *p* < 0.05 was designated as statistically significant.

## 3. Results

### 3.1. Detection and Characterization of Different Phenotypes of Circulating Tumor Cells and Circulating Cancer-Associated Fibroblasts

CTCs were not detected in 20 samples from female healthy volunteers (age range of 18–77, median of 40), whereas breast cancer cell lines of different molecular subtypes revealed the expected epithelial or EMT-related phenotypes (Appendix A). 

All samples from the 210 breast cancer patients were informative. A total of patients were analyzed prior to therapy—24 were before the subsequent line of therapy, whereas for 3 patients, those data were missing. Cells isolated from the blood were initially gated as DAPI+ and CD45/CD31−, and different sorts of investigated cells were identified within this population. Putative CTCs were detected in 58 (27.6%) of the 210 blood samples. Their numbers ranged between 1 and 459/1 mln PBMCs (median of 10/1 mln PBMCs). Among those putative CTCs, four phenotypes were identified (Figure 1A): epithelial CTCs (i.e., K+/V−), and EMT-related CTCs, such as epithelial–mesenchymal CTCs (i.e., K+/V+), mesenchymal CTCs (i.e., K−/V+), and negative for both epithelial and mesenchymal markers (i.e., K−V−; called negative) CTCs. They occurred in 30 (14.3%), 26 (12.4%), 12 (5.7%), and 21 (10%) of the 210 BC patients, respectively. Putative epithelial–mesenchymal and mesenchymal CTCs seemed to be smaller than epithelial CTCs, whereas negative CTCs were their size (Figure 1B). Epithelial–mesenchymal and negative CTCs revealed protrusions more frequently than epithelial CTCs or mesenchymal CTCs (Figure 1C). None of the CTCs expressed CD29.

In addition, cCAFs were determined simultaneously in the same samples (Figure 1A). cCAFs were not detected in 20 healthy donors. cCAFs classified as *α*-SMA+/K−/DAPI+/V±/CD45−CD31− were present only in 7 (3.3%) of the 210 breast cancer patients (range of 7–56 cCAFs/1 mln PBMCs and median of 25 cCAFs/1 mln PBMCs). They were characterized as V+ (13%) and V− (87%), and none of the detected cCAFs expressed CD29. No CD29+/*α*-SMA-/K−/DAPI+/V±/CD45−CD31− cCAFs were observed. They coincided with CTCs, in particular, epithelial and negative CTCs (R^2^ = 0.165, *p* = 0.017 and R^2^ = 0.250 = *p* < 0.001, respectively, Appendix A). They were also observed more frequently in the blood samples with higher yields of CTCs (R^2^ = 0.198, *p* = 0.004). No protrusions were detected in the cCAFs.

Of note, the numbers of the identified CTC phenotypes and cCAFs were not associated with the time between the blood sample donation and processing (range of 18–1488 min, median of 65 min), the total number of PBMCs in the analyzed sample, nor the counts of any leukocyte fractions performed in parallel during the pretreatment blood count.

Multiple clusters (range of 1–88/1 mln PBMCs) of 2-3 CTCs occurred in 8 (3.81%) patients. They consisted mainly of solely (i) epithelial, (ii) epithelial–mesenchymal, or (iii) negative CTCs (Figure 2). Heterogenous CTC clusters of mesenchymal CTCs and negative CTCs were detected in one patient (Figure 2), and in two patients, two homogenous CTC clusters of different phenotypes (epithelial and negative cells) were present. In addition, clusters of CTCs (only mesenchymal or negative ones) with normal cells (i.e., leukocytes or endothelial cells) were identified in four (1.90%) patients (Figure 2). The majority of these CTC–normal cell clusters also contained platelets (Figure 2). In all four patients, clusters of CTCs and clusters of CTCs and normal cells were co-detected. No clusters of CTCs and cCAFs were detected. 

### 3.2. Distribution of Different Phenotypes of Circulating Tumor Cells and Circulating Cancer-Associated Fibroblasts

One CTC/1 mln PBMCs was found in only 2 (1%) patients, whereas 56 (26.6%) patients had 2 or more CTCs/1 mln PBMCs (including 36 (17.1%) patients with 5 or more CTCs/1 mln PBMCs). Among the samples with 2 ≥ CTCs, 33 (58.9% of CTC-positive and 15.7% of all patients) and 23 (41.1% of CTC-positive and 11% of all patients) seeded CTCs of one phenotype or a different phenotype (i.e., homo- and heterotypic dissemination), respectively (Figure 3; for the simplification and readability of the presented graph in panel B, only patients with 5 >= CTCs are presented). Epithelial and mesenchymal phenotypes of CTCs were predominant, with epithelial–mesenchymal and negative CTCs occurring very rarely, and this effect was even more pronounced when the most enriched phenotype within the sample was considered in the CTC phenotype classification (Figure 3). 

In total, when CTC-negative and -positive cases were considered, EMT-related CTCs occurred more frequently in the luminal B/Her2+ (LumB/Her2+) BC subtype (20.4% of cases), whereas epithelial CTCs occurred in triple-negative breast cancer (TNBC) (22.4% of cases) (Chi2 test, *p* = 0.047, Appendix A). However, no statistically significant pattern of individual CTC phenotype distribution was observed when exclusive or dominant phenotype classification was considered (Figure 3A). Epithelial CTCs occurred more frequently (i.e., in 82% of 17 CTC-positive TNBC patients) and at higher numbers in TNBC, whereas mesenchymal CTCs occurred almost equally frequently among all molecular subtypes (on average, in 44.4% of CTC-positive patients). 

In general, all phenotypes of CTCs were found both in M0 and M1 patients (Appendix A). However, several detailed observations have been made. Epithelial–mesenchymal CTCs appeared only in M1 patients with LumB Her2+ breast cancer. Negative CTCs never occurred as a dominant or exclusive phenotype in M1 patients (Appendix A), while they were observed statistically more frequently in M0 patients with Her2+ tumors (both luminal and non-luminal; Fisher’s exact test, *p* = 0.016; negative CTCs were detected in 66.7% of cases with Her2+ vs. 25.7% of cases with Her2− tumors). Last but not least, epithelial CTCs were correlated with HR negativity in the whole cohort of patients (Fisher’s exact test, *p* < 0.001; epithelial CTCs were detected in 81.8% of cases with HR− vs. 34.2% of cases with HR+ tumors).

cCAFs occurred mainly in LumB/Her2+ (*n* = 3) and TNBC (*n* = 3) patients. CTC clusters occurred more frequently in the samples with higher yields of CTCs, heterotypic spread, and the presence of cCAFs, independent of the molecular subtype.

### 3.3. Clinical Relevance of Different Phenotypes of Circulating Tumor Cells and Circulating Cancer-Associated Fibroblasts

The presence, numbers, clusters, and phenotypes of CTCs and cCAFs were analyzed in relation to the patients’ clinico-pathological parameters and response to therapy in the whole cohort of patients (Table 1, Appendix A) as well as in the subcohorts of patients, defined based on the ages of the patients, their molecular subtype, N and T status, grade, and therapy timepoint. 

CTCs were detected more frequently in older patients (≥50 years old; *p* = 0.023, Table 1). All phenotypes of CTCs were detected at all stages of tumor development (cT1-4) and lymph node involvement (cN0-3), whereas cCAF (Table 1) and CTC clusters were found in ≥cT2 tumors. Negative CTCs correlated with a higher number of involved lymph nodes (Fisher–Freeman–Haltman, *p* = 0.048) (Figure 4A).

Both CTC and cCAF presence correlated with the presence of metastases (Chi2 test, *p* = 0.024 and Fisher’s test, *p* = 0.04 2, respectively) (Table 1). It seems that the presence of epithelial CTCs was particularly associated with M1 status. The frequency of metastases increased gradually, with (i) CTC-negative patients < CTC-positive patients < patients positive for both CTCs and cCAFs, and (ii) CTC-negative patients < patients positive for single CTCs < patients positive for clustered CTCs, despite the low number of cases in the last subgroups of patients in each classification (Fisher–Freeman–Haltman test, *p* = 0.006 and *p* = 0.024, respectively, Figure 4B,C). Of note, mesenchymal CTC, cCAF, and CTC clusters were observed exclusively in patients with visceral metastases (but not with specific localization) (mesenchymal CTCs: Fisher’s exact test, *p* = 0.015, Figure 4D; cCAF and CTC clusters: both *p* > 0.05), whereas epithelial CTCs occurred more frequently in patients with multiple metastatic sites (*p* > 0.05, Figure 4E), including solely visceral metastases or those coexisting with bone metastases.

None of the investigated parameters (i.e., presence, numbers, clusters, and phenotypes of CTCs, or cCAFs) correlated with the response to therapy either in the whole cohort of patients or only in patients with neoadjuvant treatment. Nevertheless, EMT-related CTCs were the most enriched in the patients with tumors that progressed during treatment (cT < pT, Chi2 test, *p* = 0.032, Figure 4F).

## 4. Discussion

Rare cancer-related cells—predominantly CTCs but also cCAFs—isolated from the peripheral blood of cancer patients are considered strong indicators of unfavorable prognosis. Here, they were investigated together in a one-tube assay using a novel method, imFC [18]. This method warranted the simultaneous detection and detailed characterization of cCAFs and both epithelial and EMT-related phenotypes of CTCs, overcoming the technical limitations of the current CTC technology. 

CTCs were enumerated at frequencies (approx. 25% and 46% of M0 and M1 BC patients, respectively) and yields (1−459/1 mln PBMCs) similar to previously published data obtained using different methodologies, including the FDA-approved “golden standard” of CTC technology, CellSearch [20,21,22,23]. Of note, we identified four, rather than typically observed three, phenotypes [24,25] related to the status of common epithelial (K) and mesenchymal (V) markers, including a K−V− phenotype. This phenotype had been, so far, putatively invisible to the current detection methods based on fluorescent microscopy, whereas the negativity of those cells for CD45, CD31, α-SMA, and CD29 (i.e., markers of leukocytes, endothelial cells, and two subpopulations of fibroblasts, respectively) as well as their size and morphology allowed us to classify them as potential CTCs. The putative K−V− CTCs were indeed the size of epithelial CTCs, revealed more frequent protrusions (not existing on cCAFs, while usually being a feature of migrating cells), occurred both in M0 and M1 patients, and correlated with Her2-positivity and a higher number of involved lymph nodes. As the EMT process is a continuum and might involve the downregulation of epithelial markers and/or the upregulation of mesenchymal markers, resulting in different intermediate stages, we assigned them to EMT-CTCs. Nevertheless, it cannot be excluded that K−V− represents another fraction of cells. Therefore, different statistical analyses were performed, including and excluding these K−V− cells from the general classification, resulting in the same correlations both for epi-CTCs and EMT-CTCs (even after the exclusion of the K−V−CTCs). Further studies, including genomic analysis, should be performed to confirm the cancer origin of these cells, understand their biology, dissect the markers allowing their unquestionable detection, and validate their clinical relevance. 

Interestingly, two other EMT-related types of CTCs, i.e., epithelial–mesenchymal and mesenchymal, seemed to be smaller than epithelial CTCs, which should be taken into account for size-based methods of CTC detection and investigated carefully in regard to the inclusion criteria for CTC classification. In total, all EMT-related CTCs appeared to correlate with some aspects of tumor progression, including the progression of tumors under systemic treatment. As EMT signatures were proven to coexist with the stemness program, it could potentially explain the resistance of those particular clones of tumor cells to therapy [26].

In the current study, we observed various patterns of tumor dissemination. Single and clustered CTCs were found, as well as CTCs clustering with normal cells of undefined type and origin. Although these clusters were identified in only 8% of patients, they were associated with a greater number of metastatic sites when compared to CTC-negative patients and patients with single CTCs. This corroborates the data showing that CTCs cluster alone and together with normal cells, e.g., neutrophils possess a higher capacity to form metastases [27,28]. As CTC clusters were seen in the samples with higher yields of CTCs, heterotypic spread, and the presence of cCAFs, it might be assumed that there is a subgroup of patients with a broad range of circulating cells, indicating putative dynamic and multifactor development of their tumors at primary or secondary sites.

Approximately 40% of the CTC-positive patient samples were characterized by heterotypic seeding, with more than one phenotype of CTCs detected. Epithelial phenotype occurred the most frequently in TNBC and in cases with multiple coexisting bone and visceral metastases. EMT-related CTCs were more frequently detected in LumB/Her2+ BC subtypes, and specifically, mesenchymal CTCs were found exclusively in cases with visceral metastases. It might be hypothesized that the mechanism of tumor dissemination is different among molecular subtypes, e.g., luminal BC might be more prone to EMT. Similarly, different secondary organs might attract or seed CTCs of different phenotypes. However, all these observations need to be verified in larger study cohorts.

cCAFs, characterized as α-SMA+/K−/V±/CD45−CD31− cells, were co-detected with CTCs, especially those of epithelial and negative phenotypes, in seven (3%) BC, mainly M1, patients. In contrast, using the same α-SMA as a marker of identification, cCAFs were previously identified by Ao et al. in 88.2% of metastatic BC patients and in 23.1% of local BC patients, whereas Ortiz-Otero et al. detected cCAFs in all metastatic BC patients [6,29]. They were also found in higher numbers per patient than in our study. One explanation for such a difference in cCAF yield might be the study cohort and methodology, as, during our study, we enrolled only 26 M1 patients, and in our protocol, the cells were stained in suspension, which might have hypothetically generated a greater loss of cells in contrast to the staining performed on microfilter [6] or glass microscope slides, as performed by the others [29]. Interestingly, the majority of the cCAFs detected in our study were negative for vimentin. In line with this observation, some studies showed that a loss of vimentin enhances fibroblasts’ motility through microchannels [30], a reduced number and size of cell–matrix adhesions [31], and cell stiffness [32], which may potentially allow them to enter the blood stream. Interestingly, no CD29+ cCAFs were found, whereas this subtype of cCAF was found predominantly within the primary tumors of luminal BC [33]. Due to the small number of cCAF-positive patients, the statistical analysis performed on those samples lacked statistical power. Nevertheless, it was noted that the frequency of metastases was greater in patients with coexisting CTCs and cCAFs in comparison to patients with no CTCs or only CTCs. cCAFs were also present in patients with exclusively visceral metastases, while, to date, there has been no study connecting cCAF presence with the type of metastases. 

Last but not least, it is important to note that, to date, only a few studies have assessed the suitability of the imFC system for CTC detection [34,35,36,37,38], and only one has been applied to BC patients, however, with a low number of cases (*n* = 8) [39]. Thus, our study is the first to use imFC on a relatively large population of patients and to focus on the multiparameter analysis of those rare cells. Of note, imFC, with its improved—in comparison to CellSearch—resolution, allows for not only defining CTCs but also examining their morphological details (e.g., protrusions), which, in the future, might even upgrade the criteria of CTC identification [40]. Multimarker analysis using imFC facilitates the application of phenotyping several subpopulations of cells (here, four phenotypes of CTCs and two putative subpopulations of fibroblasts) in one tube simultaneously and the more careful identification of CTCs based on only four exclusion markers (i.e., CD45, CD31, *α*-SMA, and CD29). In addition, owing to visual verification of the investigated cells, imFC seems to be a technology that omits the classical problems of flow cytometry in liquid biopsy. Its high-throughput analysis and standardization of acquisition parameters grant the fast documentation of a large numbers of cells in an identical manner, which warrants a kind of interpatient normalization of CTC analysis. The analysis of CTCs using imFC is certainly a completely new area of expertise. It needs to adapt to the established standards and still faces the general technical problems of CTC analysis (e.g., not having standards for positive controls: cell lines spiked into healthy blood samples do not really reflect patients’ samples). We adopted most of the rules of CTC detection workflow and criteria for their identification from CellSearch, but in the future, it might be further improved, e.g., by multicenter studies or the usage of AI tools. 

## 5. Conclusions

To sum up, to the best of our knowledge, this is the first study showing the feasibility of imFC to co-detect CTCs and cCAFs in peripheral blood collected from BC patients in a one-tube assay. Both the CTCs of different phenotypes and cCAFs might indicate more advanced disease, and might be putatively linked to dissemination to different organs, which both merit further investigation in larger cohorts of patients. 

## Figures and Tables

**Figure 1 cancers-15-04169-f001:**
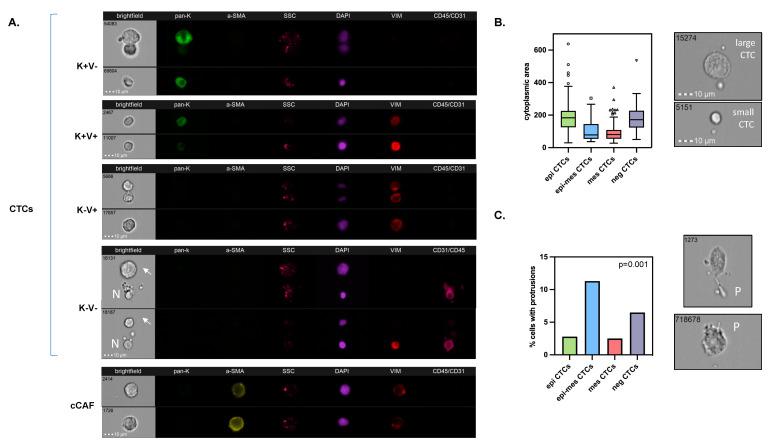
Characterization of CTCs in breast cancer: Examples of their different phenotypes (**A**), comparison of their sizes defined as cytoplasmic area (**B**), and occurrence of protrusions among them (**C**). BF indicates brightfield, pan-K—pan-keratins, a-SMA—alpha-smooth muscle actin, SSC—side scatter, Vim—vimentin, CD45/CD31—leukocyte/endothelial cell marker, N—normal cell, arrows—CTC, P—protrusions; all images were captured with objective 40×.

**Figure 2 cancers-15-04169-f002:**
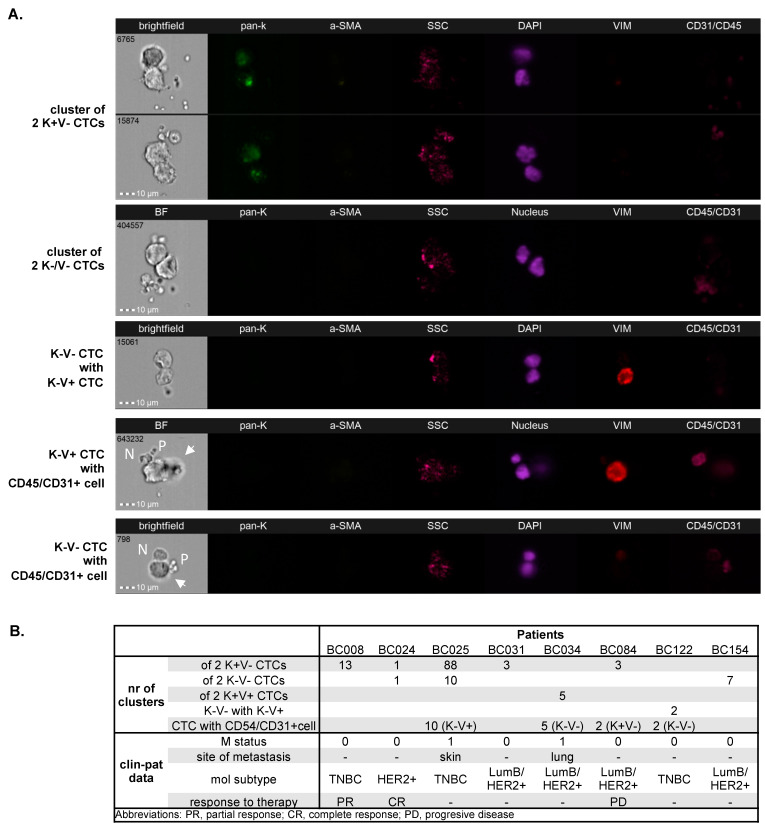
CTCs and CTC–normal cell clusters in breast cancer detected using imaging flow cytometry: Homogeneous, heterogeneous clusters, and CTC clusters with normal cells (**A**). Clinico-pathological data of cluster-positive patients (**B**). BF indicates brightfield, pan-K—pan-keratins, a-SMA—alpha-smooth muscle actin, SSC—side scatter, Vim—vimentin, CD45/CD31—leukocyte/endothelial cell marker, N—normal cell, P—platelets, arrows—CTC; all images were captured with objective 40×.

**Figure 3 cancers-15-04169-f003:**
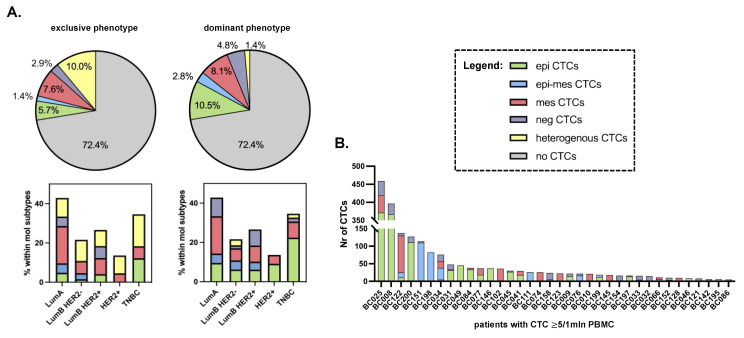
Distribution of CTC phenotypes in breast cancer patients: Distribution of exclusive and dominant CTC phenotypes in the whole cohort of patients and among different molecular subtypes of breast cancer (**A**), and heterogeneity of CTCs in individual patients with CTC ≥ 5 (**B**).

**Figure 4 cancers-15-04169-f004:**
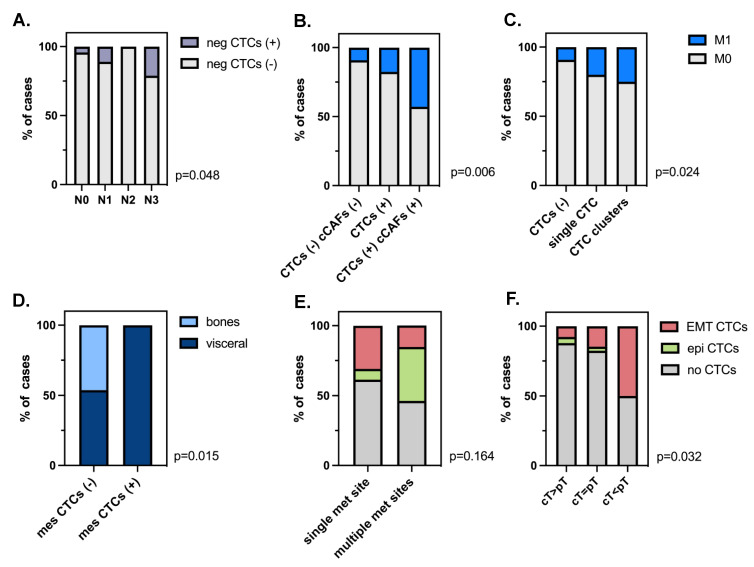
Clinical relevance of different phenotypes of CTCs and cCAFs: Distribution of negative (neg) CTCs by N status (**A**), presence of metastases depending on CTCs/cCAFs (**B**), and single/clustered CTC status (**C**); mesenchymal (mes) CTC correlation with a site of metastases (**D**), CTC phenotype correlation with number of metastases (**E**) and with response to therapy (**F**).

**Table 1 cancers-15-04169-t001:** Comparison of CTC and cCAF status to clinico-pathological features and response to treatment in breast cancer patients. F indicates Fisher’s exact test, otherwise Chi-squared test was performed. Statistically significant results are marked in bold. Due to missing data, not all numbers sum up to 210.

		CTC			cCAF		
Variable	Total *n*	CTC neg	CTC pos	*p*-Value	cCAF neg	cCAF pos	*p*-Value
**Age**							
<50	84 (40%)	68 (44.7%)	16 (27.6%)	**0.02**	83 (40.9%)	1 (14.3%)	0.25 F
≥50	126 (60%)	84 (55.3%)	42 (72.4%)		120 (59.1%)	6 (85.75)	
**cT stage**							
cT1-2	126 (71.6%)	99 (72.8%)	27 (67.5%)	0.51	125 (71.8%)	1 (50%)	0.49 F
cT3-4	50 (28.4%)	37 (27.2%)	13 (32.5%)		49 (28.2%)	1 (50%)	
**cN stage**							
cN0	70 (40.2%)	55 (41%)	15 (37.5%)	0.69	70 (40.7%)	0 (0%)	0.52 F
cN1	104 (59.8%)	79 (59%)	25 (62.5%)		102 (59.3%)	2 (100%)	
**M stage**							
M0	184 (87.6%)	138 (90.8%)	46 (79.3%)	**0.02**	180 (88.7%)	4 (57.1%)	**0.04 F**
M1	26 (12.4%)	14 (9.2%)	12 (20.7%)		23 (11.3%)	3 (42.9%)	
**Grading**							
1	11 (5.7%)	8 (5.5%)	3 (6.3%)	0.58	11 (5.8%)	0 (0%)	0.54
2	93 (48.2%)	67 (46.2%)	26 (54.2%)		90 (47.6%)	3 (75%)	
3	89 (46.1%)	70 (48.3%)	19 (39.6%)		88 (46.6%)	1 (25%)	
**Molecular type**							
Luminal A	21 (10.2%)	12 (8%)	9 (16.1%)	0.13	20 (10.1%)	1 (14.3%)	0.25
Luminal B HER2−	65 (31.6%)	51 (34%)	14 (25%)		65 (32.7%)	0 (0%)	
Luminal B HER2+	49 (23.8%)	36 (24%)	13 (23.2%)		46 (23.1%)	3 (42.9%)	
Non-luminal HER2+	22 (10.7%)	19 (12.7%)	3 (5.4%)		22 (11.1%)	0 (0%)	
Triple-negative	49 (23.8%)	32 (21.3%)	17 (30.4%)		46 (23.1%)	3 (42.9%)	
**Response to treatment**							
Stable disease	21 (51.1%)	15 (13.6%)	6 (20.7%)	0.62	21 (15.2%)	0 (0%)	0.8
Complete response	39 28.1%)	30 (27.3%)	9 (31%)		39 (28.3%)	0 (0%)	
Partial response	69 (49.6%)	56 (50.9%)	13 (44.8%)		68 (49.3%)	1 (100%)	
Progressive disease	10 (7.2%)	9 (8.2%)	1 (3.4%)		10 (7.2%)	0 (0%)	

## Data Availability

The datasets used and/or analyzed during the current study are available from the corresponding author upon reasonable request.

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
