# Peer review of "Improved Characterization of Circulating Tumor Cells and Cancer-Associated Fibroblasts in One-Tube Assay in Breast Cancer Patients Using Imaging Flow Cytometry"

_cancers, 2023, doi:10.3390/cancers15164169_

Round 1

Reviewer 1 Report

The CTC morphology is insufficiently studied. CTCs are probably morphologically different from primary tumor cells. CTCs undergo biological changes, including EMT, when entering the bloodstream. In the bloodstream CTCs are exposed to mechanical action (hydrodynamic action of blood). This can greatly affect their morphology. The CTC morphology data are undoubtedly unique and valuable for further scientific research.

This study is original. It has no signs of plagiarism and has scientific novelty. The application of imaging flow cytometry for detection of CTCs and cCAFs and their morphology analysis plays an important role in breast cancer research and it is high interesting for readers. The introduction provides sufficient background about benefits of imaging flow cytometry for detection of CTCs and cCAFs. All the cited references are relevant to the research, 48% of them for the 5 years ago. No inappropriate self-citations by authors was detect. The self-citation by authors is 10% (4 from 40 sources). The design of this study is quite considered. The research methods are well chosen, but its description requires improvement. The results need some corrections. Conclusions are supported by the results of the present study. However, in conclusions, I would also like to see the clinical relevance of detected cells.

Minor revision:

1.      Lines 104-105 – remove unnecessary paragraph.

2.      Line 145 and further - What does CD45&CD31- mean? It must be changed to CD45/CD31- or CD45-CD31-.

3.      Lines 153-154 unclear cell phenotypes. What does ?-SMA+/K-/DAPI+/V- or V+/CD45&CD31- mean? Is it ?-SMA+/K-/DAPI+/V±/CD45&CD31-? What does CD29+/K-/DAPI+/V- or V+/CD45&CD31-? Is it CD29+/K-/DAPI+/V±/CD45&CD31-?

4.      Line 218 - Were the cell clusters counted per volume or per 1 mln of leucocytes?

5.      Line 223 and further – Do normal cells mean leucocytes?

6.      Method description - MDA-MB-361 cell line should be singled out separately as LumB breast cancer metastasis into the brain. All other cell lines are primary breast tumors.

7.      Have all the cells in the samples been analyzed by imFC or only a part?

8.      Method description - There is no description of the cell cluster detection. It need to be added in the Method description.

9.      The phenotypes of exclusive and dominant CTCs should be clearly described in the Method description.

Major revision:

1.      The authors used 7 human breast cancer cell lines to optimize the CTC and cCAFs immunofluorescent staining protocol. A fibroblast and endothelial cell lines are also needed to optimize this protocol. There is no data about this cell lines in the description of the methods. It need to be added.

2.      Did the authors use Fc-block, unstained and isotypic controls? Fc-block is needed if we working with PBMCs to reduce the nonspecific antibody binding. This is especially important if we detecting very rare cells. Unstained control is needed for positive/negative fluorescence intensity cut especially in CTC detection because of its autofluorescence.

3.      Keratin, a-SMA and CD29 antibodies are also needed to optimal compensation. There is no data about these antibodies in the compensation description. It need to be added.

4.      Why do the authors consider keratin- and vimentin-negative cells to be CTCs? The negative expression of keratin and vimentin is not a generally accepted feature of CTCs, even EMT-like CTCs. The K-V- cell population may include, for example, some progenitors of bone marrow origin. These cells are also found in the blood rarely. Thus, K-V- cells don’t have any positive specific marker and can’t be considered CTCs. The authors can rename these cells to CD45/CD31- non-CTCs or exclude it from this manuscript.

I recommend accepting this manuscript after major revision.

English language and style are minor spell check required.

Reviewer 2 Report

This study focused on two types of cells found in the blood of breast cancer patients: circulating tumor cells (CTCs) and circulating cancer-associated fibroblasts (cCAFs). The researchers used a new technology called imaging flow cytometry (imFC) to simultaneously analyze these cells in the blood samples of 210 breast cancer patients. They found different types of CTCs with various characteristics related to cancer progression and discovered that cCAFs were present in some patients and were linked to the presence of metastases (cancer spread to other parts of the body). Three critical points need further clarification from the authors:

1.       The discoveries in this study have benefited from using imFC. The authors emphasized the significance of imFC in enabling simultaneous analysis of CTCs and cCAFs, overcoming previous technical limitations, while it’s not widely used to study CTCs. Hence it’s necessary to detail the working principles and advantages of imFC.

2.       Lines 139-141, authors used a sample from only one healthy donor to generate the compensation matrix, which may raise doubts about its statistical validity.  As compensation matrices are crucial in flow cytometry data analysis, the authors should provide a rationale for their selection of this specific donor and address any possible impact on the study's findings. Failing to address this issue could cast doubt on the reliability of the discoveries made using this method.

3.       The authors need to verify images shown in Figure 2 are real clusters of CTCs, instead of CTC clumping of aggregates formed during the sample processing. A thorough description of the validation method and process will add confidence to the study's results and strengthen the overall reliability of the findings.

No significant issues were found with language quality. 

Round 2

Reviewer 1 Report

I recommend to accept this manuscript in present form.

Reviewer 2 Report

The authors have added descriptions to answered the questions the reviewer previously raised. 

The quality of English is sufficient for readers to understand the content.